# Advances in Systemic Therapy for Hepatocellular Carcinoma and Future Prospects

**DOI:** 10.3390/curroncol32090490

**Published:** 2025-08-31

**Authors:** Rie Sugimoto, Miho Kurokawa, Yuki Tanaka, Takeshi Senju, Motoyuki Kohjima, Masatake Tanaka

**Affiliations:** 1Department of Hepato-Biliary-Pancreatology, NHO Kyushu Cancer Center, 3-1-1 Notame Minami-ku, Fukuoka 811-1352, Japan; miikkurom.913@gmail.com (M.K.); horikawa724@yahoo.co.jp (Y.T.);; 2Department of Gastroenterology, NHO Kyushu Medical Center, 1-8-1 Chigyohama Chūō-ku, Fukuoka 810-8563, Japan; mtaskohjima@gmail.com; 3Department of Medicine and Bioregulatory Science, Graduate School of Medical Sciences, Kyushu University, 3-1-1 Maidashi Higashi-ku, Fukuoka 812-8582, Japan; tanaka.masatake.656@m.kyushu-u.ac.jp

**Keywords:** hepatocellular carcinoma, systemic chemotherapy, combination immunotherapy, molecular targeted agent, combination of locoregional therapy and systemic chemotherapy

## Abstract

Systemic therapy for hepatocellular carcinoma is evolving rapidly. The introduction of multi-tyrosine kinase inhibitors and immunotherapy has led to improvements in response rates and survival rates. Additionally, the application of these therapies is expanding beyond advanced-stage cancer to include their combination with local therapies at the intermediate stage. Further efforts are needed to optimize treatment selection and develop new drugs to improve the prognosis of patients.

## 1. Introduction

Until around the year 2000, the treatment strategies for hepatocellular carcinoma (HCC) focused on the surveillance of high-risk cases with hepatitis C or hepatitis B for its early detection, with local therapy being the primary treatment modality. Advances in systemic therapy progressed little after the introduction of the multiple tyrosine kinase inhibitor (mTKI) sorafenib in 2008 as a first-line treatment for HCC [1]. However, with the advent of nucleoside analogues that can cure hepatitis C, the underlying risk factors for HCC have altered, shifting the primary cause of carcinogenesis from viral hepatitis to metabolic dysfunction-associated steatohepatitis, including non-alcoholic steatohepatitis [2]. This made it difficult to identify high-risk cases.

Another mTKI, lenvatinib, demonstrated non-inferior overall survival (OS) compared with sorafenib and had favourable progression-free survival (PFS) and objective response rates (ORR) as a first-line therapy [3]. Second-line treatment with mTKIs, such as regorafenib [4], emerged following disease progression in patients on sorafenib. Other second-line therapies, e.g., ramucirumab [5] and cabozantinib [6], were subsequently introduced as effective treatment options. Furthermore, combination immunotherapy has been added to these options. Atezolizumab plus bevacizumab is the first combination immunotherapy for HCC and the first regimen to demonstrate superior OS and PFS compared with sorafenib [7]. Durvalumab plus tremelimumab, the first regimen to combine two immune checkpoint inhibitors (ICIs), had superior OS compared with sorafenib [8]. Furthermore, nivolumab plus ipilimumab demonstrated superior OS compared with lenvatinib or sorafenib [9].

Drug therapy was initially indicated for advanced cancer where local therapies such as resection or transarterial chemoembolization (TACE) were not feasible, and was primarily targeted at Barcelona Clinic Liver Cancer (BCLC) stage C. However, the concepts of TACE-refractory [10] and TACE-ineligible [11] patients emerged, significantly expanding the treatment population. Although local therapy was initially thought to have a narrower indication, its application has expanded through the combination of various systemic drug therapies [12,13]. The era has arrived where local therapy and drug therapy are not mutually exclusive but can be combined to improve patient prognosis. Furthermore, with the advancement of precision medicine, we are approaching treatment modalities in which drug therapy using point mutations in cancer genes can be applied to liver cancer. This review discusses the advances in various drug therapies for HCC, prospects for the development of new therapeutic agents, and strategies for combining and optimizing treatment modalities. Herein, advanced HCC is defined as cases with BCLC stage B or C and Child–Pugh class A, for which curative treatments such as ablation therapy, TACE monotherapy, or resection are not expected to be beneficial, and for which systemic drug therapy is indicated. These cases were included in the scope of this review.

## 2. Advances and Prospects in Systemic Therapy for Advanced-Stage HCC

### 2.1. Advances and Prospects in First-Line Therapies

In 2008, the SHARP trial demonstrated that sorafenib significantly outperformed placebo in terms of OS and PFS for the treatment of advanced HCC, making it the first oral mTKI approved as a systemic drug therapy for HCC [1]. The median OS of sorafenib was 10.7 months compared with 7.9 months for placebo. Subsequent trials failed to demonstrate the superior efficacy of sunitinib [14], linifanib [15], and brivanib [16] over sorafenib, but in 2018, the REFLECT trial finally demonstrated that lenvatinib, an oral mTKI with selective inhibitory activity against receptor tyrosine kinases including vascular endothelial growth factor (VEGF) receptor, fibroblast growth factor receptor, platelet-derived growth factor receptor, KIT, and RET, was non-inferior to sorafenib [3]. The median OS of patients treated with lenvatinib was 13.6 months compared with 12.3 months for sorafenib. The overall response rate (ORR) in the REFLECT trial was 40.6% (sorafenib 12.4%, odds ratio 5.01). Because of this high response rate, lenvatinib became a key drug in subsequent combination trials. A key feature of lenvatinib is its high tumour blood flow inhibitory effect, characterized by early tumour blood flow reduction and tumour vascular normalization [17]. Although a side effect of lenvatinib is fatigue, various administration methods have been developed to mitigate this while maintaining treatment intensity, including the weekend-off regimen, where treatment is suspended on weekends [18].

ICIs have demonstrated promising therapeutic effects in various types of cancer; however, nivolumab monotherapy for HCC failed to demonstrate superiority over sorafenib [19]. This is thought to be related to the fact that HCC often presents in a state where the immune response is not ‘hot,’ as discussed later. To address this, efforts have been made to develop combination therapies combining molecular-targeted drugs with ICIs to induce a ‘hot’ immune response. In 2020, groundbreaking trial results were reported. In the IMbrave150 trial, atezolizumab (anti-programmed cell death ligand-1 [PD-L1] antibody) plus bevacizumab (anti-VEGF antibody) significantly extended OS compared with sorafenib (19.2 months vs. 13.4 months) [20] and significantly improved PFS (6.9 months vs. 4.3 months) [7]. The hazard ratio (HR) was 0.66 (95% confidence interval [CI] 0.53–0.85) for OS and 0.65 (95% CI 0.53–0.81) for PFS, both of which were statistically significant. Thus, atezolizumab plus bevacizumab was the first immunoconjugate therapy to demonstrate efficacy for the treatment of HCCs, and it was positioned as a first-line therapy.

Cancer cells can be classified as “immune hot” (high infiltration of CD8+ and CD3+ T cells in colon cancer tissues), “immune cold” (low immune cell infiltration), both of which were closely associated with survival rates [21], and “immune altered” (excluded) [T cells present in the tumour periphery cannot enter the tumour, an effect strongly related to VEGF expression [22]]. If the immune system is functioning normally, “immune hot” tumours should be eliminated by the immune system, but if not, this may result in carcinogenesis. This suggests that the programmed cell death protein-1 (PD-1)/PD-L1 immune checkpoint mechanism [23] is involved at the tumour site, and therefore, treatment with PD-1/PD-L1 inhibitors alone should be effective. However, only about 30% of HCCs show lymphocyte infiltration of the tumour (“immune hot”), about 30% have WNF/β-catenin mutations without CD8+ cell infiltration (“immune altered [excluded]”), and in some cases, T-cell activity is suppressed by the immunosuppressive microenvironment (“immune cold”) [24].

Atezolizumab plus bevacizumab therapy is thought to exert its antitumour effect through the combined effects of an ICI plus an anti-VEGF inhibitor, by suppressing immunosuppressive Tregs, restoring the immune activity of CD8+ cells [25], normalizing the tumour vasculature, promoting CD8+ cell infiltration into the tumour [26], and inducing cell death and subsequent cancer antigen release through direct antitumour and necrotic effects [27,28]. The most frequent adverse reactions reported as grade 3 or higher were hypertension (12.2%), increased aspartate aminotransferase (5.1%), increased alanine aminotransferase (1.3%), thrombocytopenia (1.3%), and proteinuria (0.6%). The study also included patients with a portal vein tumour plug (Vp)4, demonstrating efficacy in patients with or without a tumour plug [7]. Additionally, a Phase II study that was conducted to evaluate the efficacy and safety of this regimen in Child–Pugh B patients showed a certain level of efficacy and safety [29]; however, the number of patients was low (*n* = 31), and therefore, it did not provide definitive evidence.

Durvalumab (anti-PDL-1 antibody) plus tremelimumab (anti-cytotoxic T-lymphocyte-associated protein 4 [CTLA-4] antibody), the first combination regimen of two ICIs, significantly improved OS compared with sorafenib in the HIMALAYA trial (16.4 months vs. 13.8 months [HR 0.78, 95% CI 0.65–0.92]) [8]. Additionally, an HR of 0.76 for OS at 5 years was reported. This regimen is the first VEGF inhibitor-free regimen for HCC and can be used in patients with bleeding risk, urinary protein, refractory hypertension, and thromboembolism risk. Given the increasing age of patients diagnosed with HCC, and the fact that viral cirrhosis is decreasing and metabolic dysfunction-associated steatohepatitis is increasing as a cause of liver cancer, the indications for VEGF-free regimens may increase. The frequency of immune-mediated adverse events is relatively high, and the use of high-volume systemic steroids at a rate of 20.1% should be noted. The HIMALAYA trial also tested durvalumab alone, which was non-inferior to sorafenib in OS [8]. This VEGF inhibitor-free regimen had a low rate of grade ≥3 adverse events (12.9%), making it a relatively safe drug to use without the risk of urinary protein in an ageing population of liver cancer patients.

In May 2025, the results of the CheckMate9DW study of nivolumab (anti-PD-1 antibody) plus ipilimumab (anti-CTLA-4 antibody) therapy were published [9], indicating a significantly improved OS compared with lenvatinib or sorafenib (23.7 months vs. 20.6 months [HR 0.79 95% CI 0.65–0.96]). Kaplan–Meier curves for this regimen crossed early, with a survival advantage for lenvatinib or sorafenib in the first 6 months, but a consistently better OS for nivolumab plus ipilimumab thereafter (HR 1.65 95% CI 1.12–2.43 for nivolumab plus ipilimumab at 6 months). This was the first study to demonstrate efficacy against a control arm that included lenvatinib, whereas all previous studies that showed efficacy had used sorafenib as the control arm. The study excluded Vp4, so caution should be exercised when interpreting the results. The ORR for nivolumab plus ipilimumab was 36%, which was significantly better than the control arm (13%) (*p* < 0.0001). Immunological adverse events with nivolumab plus ipilimumab occurred in 58% of patients, with grade 3 and 4 events occurring in 28% of patients. High-volume steroids were required in 29% of patients and treatment-related deaths occurred in 12 patients (3%), most of which were related to liver injury. Therefore, the liver function of patients should be closely monitored in the future.

In 2025, the results of the HEPATORCH trial were published [30], showing that toripalimab (anti-PD-1 antibody) plus bevacizumab was a significant improvement over sorafenib with a PFS of 5.8 months vs. 4.0 months (HR 0.69, *p* = 0.0086; OS: 20.0 months vs. 14.6 months; HR 0.76, *p* = 0.039), which has led to its approval by China’s State Administration of Medicines.

In the CheckMate 459 single-agent trial, nivolumab was not superior to sorafenib with an OS of 16.4 months vs. 14.7 months [19]. However, a Phase I/II study in Child–Pugh B patients (CheckMate 040 cohort 5) showed efficacy of 12% for the ORR and 55% for disease control rate (DCR), with a good safety profile, indicating the potential of nivolumab for the unmet clinical needs of Child–Pugh B patients [31].

Pembrolizumab was effective in the Phase II KEYNOTE-244 single-agent trial, with an ORR of 16% and a median OS of 17 months [32]. However, in the LEAP-002 trial evaluating combination immunotherapy, pembrolizumab plus lenvatinib had a median OS of 21.2 months, but it failed to meet the pre-specified *p*-value for lenvatinib and did not demonstrate efficacy [33]. In the REFLECT trial, which compared lenvatinib monotherapy with sorafenib monotherapy, the OS for lenvatinib monotherapy was 13.6 months. However, as the use of lenvatinib has improved in clinical practice, the OS for lenvatinib monotherapy has been extended. In the LEAP-002 trial, the OS of the control arm (lenvatinib monotherapy) increased to 19 months, resulting in a smaller difference compared with the OS of lenvatinib plus pembrolizumab (21.2 months), and thus the initially anticipated HR was not met. As seen for existing treatments, improvements in patient prognosis related to the development of various treatment sequences have been observed with other agents. In the SHARP trial, the median OS of the sorafenib group was 10.7 months, whereas in the REFLECT trial, the median survival duration was 13.6 months for lenvatinib and 12.3 months for sorafenib. However, in the IMbrave150 trial, the median OS was 13.4 months for sorafenib, while it was 20.6 months for both sorafenib and lenvatinib in the CheckMate 9DW trial, indicating that the OS is improving with sorafenib. Overall, these studies suggest the prognosis for HCC patients is steadily improving.

### 2.2. Recent Developments in First-Line Therapy for HCC

Numerous trials aimed at developing first-line drugs for future use are currently underway. Among the trials with published results, in addition to the previously mentioned LEAP-002 trial, the COSMIC 312 trial compared cabozantinib plus atezolizumab with sorafenib. An interim analysis failed to demonstrate an improvement in the OS (15.4 months for cabozantinib plus atezolizumab vs. 15.5 months for sorafenib), but the PFS was significantly improved (6.8 months vs. 4.2 months; HR 0.63, *p* = 0.0012). Therefore, trials are continuing because these treatments may be beneficial for specific patient populations [34].

The anti-PD-1 antibody tislelizumab demonstrated non-inferior OS prolongation compared with sorafenib in the RATIONALE-301 trial, with a median OS of 15.9 months vs. 14.1 months (HR 0.85), and an objective response rate of 14.3% vs. 5.4% [35]. Additionally, grade ≥ 3 adverse events were less frequent compared with sorafenib (22.2% vs. 53.4%).

There are also several ongoing Phase II/III clinical trials (Table 1). The IMbrave152 trial is comparing combination therapy consisting of atezolizumab and bevacizumab plus tiragolumab with the combination therapy of atezolizumab and bevacizumab alone (NCT05904886) [36]. A similar additional therapy trial, TRIPLET-HCC (PRODIGE81-FFCD2101) [37] (NCT05665348), is assessing the addition of ipilimumab to atezolizumab and bevacizumab and is ongoing worldwide. 

Regarding the development of drugs with mechanisms of action other than immune checkpoint inhibition, a Phase II/III trial is currently underway comparing atezolizumab plus bevacizumab and tremelimumab plus durvalumab with the TGF-β inhibitor livmoniplimab and the PD-1 antibody budigalimab (NCT06109272) [38]. The SIERRA trial is targeting the unmet clinical needs of Child–Pugh B patients, and a trial administering tremelimumab plus durvalumab to patients with HCC scoring 7 or 8 on the Child–Pugh B scale is also ongoing (NCT05883644) [39]. A global Phase III trial comparing the combination of a new IgG4-targeting PD-1 monoclonal antibody, nofazinlimab, plus lenvatinib to lenvatinib alone is currently underway. This combination therapy demonstrated favourable results in a Phase Ib trial conducted in China, with an ORR of 45% and PFS of 10.4 months [40]. Another clinical trial is comparing a combination of anti-CTLA-4 antibody and anti-PD-1 antibody with sorafenib (NCT04720716). The primary endpoints are OS and ORR [41]. The APOLLO trial, conducted in China, is comparing the combination therapy of anlotinib, an anti-angiogenic drug targeting multiple receptor tyrosine kinases including VEGFR2 and VEGFR3, plus the anti-PD-1 antibody penpulimab, with sorafenib. A second interim analysis conducted in 2025 reported a median PFS of 6.9 months vs. 2.8 months, with a median OS of 13.2 months vs. 16.5 months, demonstrating a statistically significant improvement compared with sorafenib [42]. A Phase III trial (NCT05408221) is comparing a combination of the anti-PD-1 IgG1 monoclonal antibody rulonilimab plus lenvatinib vs. lenvatinib plus placebo [43].

The NCT04401800 Phase II trial conducted in China, a single-arm study adding tislelizumab which demonstrated efficacy as a single agent for advanced HCC, to lenvatinib, reported an ORR and DCR of 38.7% and 90.3%, respectively, with a median PFS of 8.2 months [44]. The RENOBATE Phase II trial reported an ORR of 21% for regorafenib plus nivolumab, a median PFS of 7.3 months and a 1-year OS of 80.5% (NCT04310709) [45]. The NCT05924997 Phase Ib/II trial is evaluating the efficacy and safety of adebrelimab, camrelizumab, and apatinib as a first-line treatment [46].

### 2.3. Advances and Prospects in Second-Line and Beyond Therapies

To date, following the failure of sorafenib, the only agents with randomized prospective trial evidence as second-line or later therapies are regorafenib (RESORCE trial) [4], cabozantinib (CELESTIAL trial) [6], and the monoclonal antibody ramucirumab (REACH-2 trial, for cases with α-fetoprotein [AFP] levels ≥400 ng/mL) [5]. Regorafenib demonstrated a significant improvement in OS and PFS for patients who were resistant to sorafenib and who tolerated the treatment in the RESORCE trial [4]. Ramucirumab, an antibody targeting VEGFR2, demonstrated efficacy in the REACH-2 trial targeting patients with AFP ≥400 ng/mL [5]. Furthermore, a subgroup analysis showed that the same intensity of treatment was possible, and similar outcomes were achieved in older patients when patients were divided into those younger than 65 years, those aged between 65 and 75 years, and those older than 75 years [47]. Cabozantinib has an inhibitory mechanism distinct from other molecular-targeted agents, including VEGFR2, MET, and AXL. The CELESTIAL trial demonstrated improvements in OS and PFS [6].

Pembrolizumab demonstrated a statistically non-significant improvement in OS of 13.9 months vs. 10.6 months (HR 0.781, 95% CI: 0.611–0.998, *p* = 0.0238) compared with the placebo in patients with HCC who had previously received sorafenib therapy (KEYNOTE-240 trial) [48]. However, this regimen may represent a useful option for patients who have failed mTKI therapy.

There are currently no treatment regimens with evidence supporting their use as second-line therapy following ICIs. However, in the IMbrave150 trial, the rate of progression to second-line therapy was 35% [49], indicating that the efficacy of second-line therapy should be carefully considered.

Currently, the IMbrave251 trial (NCT04770896) is assessing secondary treatment regimens in patients who failed to demonstrate efficacy after treatment with ICIs. This trial is comparing the combination of atezolizumab plus lenvatinib or sorafenib with lenvatinib or sorafenib alone in patients who have failed treatment with atezolizumab plus bevacizumab [50]. Additionally, the ongoing LIVERATION trial (NCT05201404) is evaluating the efficacy and safety of namodenoson (an A3 adenosine receptor agonist) in patients with Child–Pugh B7 HCC who experienced disease progression after first-line therapy [51]. Furthermore, the results of the efficacy and safety of regorafenib plus pembrolizumab following ICI treatment have been published in a Phase II study, with a median PFS of 2.8 months after atezolizumab plus bevacizumab and 4.2 months after other ICI treatments (NCT04696055) [52].

A Phase II study of TQB2450, an anti-PD-1 IgG1 monoclonal antibody, in combination with anlotinib, is also ongoing in patients with HCC who have failed ICI therapy (FAITH trial; NCT06031480) [53]. A Phase II trial administering suvemcitug, which inhibits VEGFR1 and 2, and the anti-PD-L1 antibody envafolimab, was conducted in patients with HCC who had received prior treatment. An interim analysis revealed an ORR of approximately 11.1% and a PFS of approximately 4.3 months [54]. Additionally, a Phase II trial is ongoing in patients with HCC who previously received ICI therapy, randomizing them to receive lenvatinib or sorafenib in combination with zabadinostat (CXD101) and geptanolimab [55]. Another Phase II trial in patients with HCC who previously received lenvatinib treatment reported a median PFS of 9.70 months and a median OS of 17.23 months. These results suggest that the combination of atezolizumab plus bevacizumab may be a treatment option for patients with advanced HCC who previously received lenvatinib treatment (jRCT1041200068) [56]. Table 2 shows a list of Phase II/III trials currently underway for the development of secondary treatments for hepatocellular carcinoma.

### 2.4. Treatment Algorithms for Advanced HCC in the Guidelines of Various Countries (As of May 2025)

Although guidelines have been published by various countries worldwide, this review will discuss the drug therapy algorithms for advanced HCC as outlined in the representative guidelines of the European Society for Medical Oncology (ESMO), European Association for the Study of the Liver (EASL), the American Association for the Study of Liver Diseases (AASLD), and the Japanese Society of Hepatology (JSH).

The most recently revised guidelines are those of the ESMO, which were published in 2025 [57]. If immunotherapy is indicated as the first-line treatment for BCLC B-C, the following therapies are recommended: atezolizumab plus bevacizumab, durvalumab plus tremelimumab, camrelizumab plus rivoceranib, nivolumab plus ipilimumab, durvalumab, or tislelizumab. If immunotherapy is not indicated, lenvatinib or sorafenib is indicated. For second-line therapy following the failure of ICI therapy, lenvatinib, regorafenib, cabozantinib, sorafenib, or ramucirumab (AFP ≥ 400 ng/mL) are recommended for grade IV. Where ICI is not appropriate and lenvatinib has failed, sorafenib, regorafenib, cabozantinib, and ramucirumab (AFP ≥ 400 ng/mL) are listed as recommendations for grade IV. If sorafenib has failed, regorafenib, cabozantinib, and ramucirumab (AFP ≥ 400 ng/mL) are listed as recommendations for grade I.

The EASL guidelines discuss various drug therapies in the section on systemic therapies. For patients with advanced HCC, preserved liver function (Child–Pugh A), and Eastern Cooperative Oncology Group (ECOG) performance status of 0–1, they recommend combinations including at least one PD-1 or PD-L1 inhibitor, provided there are no contraindications (Level of Evidence 1, strong recommendation, strong consensus). Regarding the choice of drugs, combination therapy containing PD-1 or PD-L1 inhibitors should be considered for the first-line standard of care for those without contraindications to ICIs (and bevacizumab). There is no evidence to support the use of one option in preference to another. As a treatment-switching strategy, mTKIs are considered a second-line option if progression is confirmed after ICI plus VEGF combination therapy. In cases of Child–Pugh B or higher, or risk of liver failure, ICIs are noted to be used with caution. There are no clear differences in treatment selection based on the etiology of HCC, and no overall effect on treatment efficacy has been observed. Furthermore, the guidelines emphasize the importance of multidisciplinary team-based patient-centred decision-making and urge optimal choices at each stage of treatment and diagnosis [58]. 

According to the AASLD recommendations [59], the first step for patients with BCLC C or BCLC B with multifocal disease, contraindications for local therapy or progression after local therapy, Child–Pugh A, or ECOG 0 or 1, is to determine whether immunotherapy is appropriate. If so, the risk of gastrointestinal bleeding should be assessed. If the risk is low, atezolizumab plus bevacizumab should be considered. If there is a risk for bleeding, durvalumab plus tremelimumab or sorafenib or lenvatinib is recommended. If immunotherapy is not appropriate, sorafenib or lenvatinib is recommended. For second-line therapy, if progression occurs after atezolizumab plus bevacizumab, sorafenib or lenvatinib is primarily recommended, with regorafenib, cabozantinib, ramucirumab, and durvalumab plus tremelimumab weakly recommended. Following the failure of sorafenib or lenvatinib, the administration of regorafenib, cabozantinib, or ramucirumab is strongly recommended, with nivolumab plus ipilimumab or pembrolizumab recommended with caution.

According to the JSH guidelines [60], the applicability of combination immunotherapy for HCC cases with good performance status and Child–Pugh class A that are not eligible for surgical resection, liver transplantation, or local therapy such as TACE, is determined based on the presence or absence of the condition. If applicable, atezolizumab plus bevacizumab or durvalumab plus tremelimumab is recommended. If not, sorafenib, lenvatinib, or durvalumab are listed as alternatives. A notable feature is that two immunotherapy regimens can be used sequentially as second-line therapy. Additionally, a comparison table of results from the IMbrave150 trial and the HIMALAYA trial is provided as an explanation of this algorithm, noting that the HRs for OS relative to sorafenib were 0.65 and 0.78, respectively, and differences in steroid usage rates were also clearly observed [61].

As such, treatment sequences and drug therapy usage differentiation for HCC vary slightly between the guidelines of different countries, and there is currently no clear evidence for any particular sequence.

Drugs for treating advanced HCC are broadly classified into molecular-targeted drugs centred around VEGF inhibitors and ICIs, and the development of treatments has mainly focused on combinations of two or three of these drug types. Currently, the combination of two or three ICIs with different mechanisms of action (anti-PD-1 antibodies/anti-PD-L1 antibodies/anti-CTLA-4 antibodies) or the combination of a molecular-targeted drug with any ICI is the mainstream treatment for advanced HCC. For “immune hot” tumours, monotherapy with anti-PD-1 antibodies/anti-PD-L1 antibodies is expected to be effective. For “immune altered (excluded)” tumours, a combination of anti-VEGF antibodies and anti-PD-1 antibodies/anti-PD-L1 antibodies is likely to be effective. For “immune cold” tumours, a combination of anti-PD-1 antibodies/anti-PD-L1 antibodies and anti-CTLA-4 antibodies is likely to be effective. Additionally, a combination of ICIs was reported to be beneficial in some HCC cases, with effects lasting over a long period [62].

However, various immune-related adverse events have been reported in patients treated with drug combinations. Even when using the same ICIs, immune-related adverse events may differ depending on the type of ICI [9,30,63]. Whether patients can tolerate such adverse events will be an important consideration when establishing a treatment strategy. Given the variety of treatment strategies available, it is extremely important to carefully assess each patient’s tumour status, liver reserve capacity, social background, and overall condition when developing a personalized treatment strategy.

## 3. Advances and Prospects for the Use and Combination of Drug Therapy and Locoregional Therapy for Intermediate- and Advanced-Stage HCC

### 3.1. Differentiation Between Drug Therapy and Local Therapy

Unlike other cancers, liver cell cancer has several local treatment options, such as ablation therapy and TACE. Therefore, local therapy has traditionally been considered the primary treatment for middle-stage cancer. In particular, TACE has historically been considered appropriate if the cancer is confined to the liver. TACE provides long OS when a complete response (CR) is achieved with initial treatment, but the prognosis is poor when a CR is not achieved [64]. TACE non-response is defined as the absence of efficacy after two consecutive TACE procedures or the occurrence of vascular invasion or extrahepatic metastasis. In such patients, it has been suggested that switching to systemic drug therapy rather than continuing TACE may improve their prognosis. However, it was reported that 20–30% of patients who continue treatment until TACE non-response occurs, progress to Child–Pugh B [65]. As a result, a deterioration in liver reserve function may prevent subsequent systemic drug therapy. In response to this, the concept of TACE unsuitability was proposed. TACE unsuitability can be defined as follows.

(i)Unlikely to respond to TACE: confluent multinodular type, massive or infiltrative type, simple nodular type with extranodular growth, poorly differentiated type, intrahepatic multiple disseminated nodules, or sarcomatous changes after TACE.(ii)Likely to develop TACE failure/refractoriness: Up-to-7 criteria out nodules.(iii)Likely to become Child–Pugh B or C after TACE: Up-to-7 criteria out nodules (especially bilobar multifocal HCC) modified albumin–bilirubin grade 2b.

The Up-to-7 criteria is a classification standard based on whether the sum of the tumour size and number exceeds 7, reflecting the tumour volume. A retrospective analysis by Kudo et al. compared groups that received lenvatinib therapy and those that underwent TACE in a propensity-matched control analysis of a group where TACE was expected to be ineffective and reported that lenvatinib was more effective in terms of OS and PFS [66]. Although it was a retrospective study and bias was unavoidable, thus requiring careful interpretation, it contains very important implications. In addition to this definition, other criteria such as Up-to-11 also exist as factors to determine the efficacy of TACE [67].

Regarding the future prospects of TACE and systemic drug therapy, the ABC-HCC trial (NCT04803994) [68] is currently underway. This is a global Phase IIIb randomized, multicentre, open-label trial comparing atezolizumab plus bevacizumab with TACE for intermediate-stage HCC, and the results are eagerly awaited. Additionally, the REPLACE trial (NCT04777851) [69] comparing regorafenib plus pembrolizumab with TACE is also underway.

### 3.2. Combination of Drug Therapy and Local Therapy

In 2019, Kudo et al. reported that sorafenib plus TACE significantly improved PFS compared with TACE alone [70], in contrast to previous combination trials of sorafenib and TACE that did not demonstrate efficacy [71]. One reason for this was the use of the TACE-specific treatment efficacy assessment (RECICL) to highlight the efficacy of TACE as a local therapy. Although this trial did not show a statistically significant difference in OS, it was considered clinically useful, and subsequent reports continued to describe combinations of drug therapy and local therapy. The TACTICS-L trial reported that a combination of lenvatinib plus TACE achieved a PFS of 28 months and an ORR of 88.7% in patients with BCLC B HCC [12]. Peng et al. reported the results of a randomized trial comparing lenvatinib monotherapy with lenvatinib plus TACE for advanced HCC; in this study, lenvatinib plus TACE demonstrated significantly better overall OS [13].

In 2023, atezolizumab plus bevacizumab followed by curative conversion (ABC conversion) therapy combining atezolizumab plus bevacizumab therapy with TACE, radiofrequency ablation, and resection was reported. This treatment achieved a CR rate of 30% and a complete remission (cancer-free, drug-free) rate of 19% [72]. Additionally, in 2024, it was reported that durvalumab plus bevacizumab plus TACE significantly improved PFS compared with TACE alone (10.5 months vs. 8.2 months, HR 0.77; Phase III, EMERALD-1 trial; NCT03778957) [73]. However, in that trial, no additive effect of durvalumab alone over TACE was observed, and the efficacy of combination therapy with TACE using ICI alone without VEGF inhibition was not established. 

A large-scale randomized controlled clinical trial is currently underway to evaluate the combination of atezolizumab plus bevacizumab therapy with TACE for unresectable HCC (IMPACT trial jRCTs051230037) [74], and the results are awaited. Other trials currently underway include the TALETACE study, a randomized trial comparing TACE monotherapy with atezolizumab plus bevacizumab therapy followed by TACE in patients with TACE tumours whose maximum diameter plus tumour count is six or more, and who are relatively close to the Up-to-7 criteria [75] or TACE monotherapy. The EMERALD-3 trial is a study comparing TACE alone with TACE plus durvalumab plus tremelimumab therapy with or without lenvatinib [76]. Other trials include the LEAP-012 global trial (lenvatinib plus pembrolizumab plus TACE vs. TACE alone) [77] and the CheckMate 74W trial (TACE plus nivolumab plus ipilimumab vs. TACE plus nivolumab vs. TACE alone) [78]. In China, a Phase II trial combining camrelizumab plus lenvatinib with raltitrexed and oxaliplatin (RALOX) as hepatic arterial infusion chemotherapy (HAIC) is underway, with a median PFS of 13.8 months (95% CI, 8.8–20.5) based on RECIST v1.1. The 6-month and 12-month OS rates were 94.3% and 64.9%, respectively, suggesting promising results to come [79]. Phase II/III trials comparing ongoing TACE with TACE in combination with mTKIs and/or ICIs are shown in Table 3.

## 4. Systemic Therapy and Radical Treatment

Conventionally, drug therapy has been indicated only for HCC when resection is not feasible. However, reports have increasingly suggested that conversion resection may be possible after drug therapy [80], leading to the need for a definition of “borderline resectable” in relation to resection. In 2023, the JSH and the Japanese Society of Hepato-Biliary-Pancreatic Surgery published an expert consensus statement [81] where the resection feasibility classification does not include an “unresectable” category; instead, it divided cases into three categories: resectable, borderline 1, and borderline 2. This classification was adopted because the emergence of potent chemotherapy has made resection possible in cases where upfront resection was previously considered impossible because of portal vein tumour thrombus or distant metastasis. Regarding preoperative chemotherapy, the LENS-HCC study [82] demonstrated the efficacy of lenvatinib as preoperative adjuvant chemotherapy, and the RACB trial [83] is currently underway to evaluate the feasibility of conversion resection using atezolizumab plus bevacizumab.

## 5. Advances and Prospects of Drug Therapy as Adjuvant Chemotherapy After Surgery

Regarding postoperative chemotherapy for HCC, an interim analysis of the IMbrave050 trial reported that the combination of atezolizumab plus bevacizumab significantly improved PFS in patients with high-risk HCC following surgery [84]; however, the final analysis did not demonstrate efficacy [85]. Currently, there are no proven treatment effects of postoperative adjuvant chemotherapy, but multiple trials are ongoing. The CheckMate 9DX trial (NCT03383458) is evaluating the efficacy of nivolumab for PFS in patients with HCC at high risk for recurrence after resection or ablation [86]. The EMERALD-2 trial (NCT03847428) [87] is evaluating the efficacy and safety of durvalumab monotherapy and durvalumab plus bevacizumab combination therapy as adjuvant therapy for patients at high risk for recurrence following curative resection or ablation. The KEYNOTE-937 (NCT03867084) trial is evaluating the safety and efficacy of pembrolizumab as adjuvant therapy for patients with HCC who achieved a radiological CR following surgical resection or local ablation, compared with placebo [88]. The results of these trials are awaited. Additionally, there have been reports of randomized double-blind trials using traditional Chinese medicine for the prevention of tumour recurrence after liver resection, with efficacy reported for multiple drugs [89,90]. Further verification will be necessary in the future.

## 6. Differences in the Efficacy of Drug Therapy Based on HCC Histological Subtypes and Gene Expression

HCC is histologically and pathologically highly diverse, and several subtypes have been suggested to influence the prognosis and treatment response. However, there are currently no high-level randomized trials demonstrating differences in treatment efficacy associated with HCC subtype. A pathological subtype termed macrotrabecular-massive (MTM) HCC is defined by the presence of predominant (>50%) macrotrabecular architecture (more than six cells thick) and is associated with cancer malignancy [91]. MTM-HCC has been reported to be strongly associated with high recurrence and metastasis rates after radiofrequency ablation [92] or surgery [93], as well as poor survival rates. However, there are currently no high-level reports demonstrating a clear correlation between the efficacy of drug therapy and MTM-HCC. Gene expression in HCC involves TERT promoter mutations and Wnt/β-catenin mutations [94], and the disease can be classified into three subtypes: (i) mitogenic and stem cell-like tumours with chromosomal instability; (ii) *CTNNB1*-mutated tumours displaying immune suppression; and (iii) metabolic disease-associated tumours, which include an immunogenic subgroup characterized by macrophage infiltration and a favourable prognosis [95]. The REFLECT trial, studying a subset determined using tumour tissues, reported that Wnt/β-catenin pathway and DNA repair pathway activation clusters in the lenvatinib group had a significant association with OS [96]. Therefore, there is potential for these findings to be developed into biomarkers for prognosis prediction and treatment response at the molecular level. However, these are all retrospective studies, and further research is needed.

## 7. Future Directions of HCC Chemotherapy

Currently, various drug development efforts are underway for HCC at different stages, including multiple combination immunotherapies and novel molecular-targeted agents for advanced HCC, combinations of TACE with molecular-targeted agents for HCC classified as BCLC B, combinations of TACE with combination immunotherapy, combinations with radiation therapy, and chemotherapy aimed at preventing tumour recurrence. At present, the prognosis for liver cancer is still poor, but future drug therapy for HCC is expected to shift from a treatment approach focused on extending survival by sequentially administering single agents to a treatment approach aimed at achieving a cure by combining various drugs and local therapies. Although further long-term survival is expected, challenges remain regarding how to optimize the use of various drugs, including their selection and sequence of administration, and the development of appropriate biomarkers that include genetic mutations and histological characteristics for optimal drug use. Regarding the treatment of HCC, treatment sequences based on evidence such as gene mutations, which have been proposed for other cancers, have not yet been developed. In the future, it is likely that treatment development, including individualized treatment, will be necessary.

## 8. Conclusions

Systemic therapy for HCC has evolved rapidly over the past decade. Currently, many Phase III trials are underway. In clinical practice, it is essential to correctly understand these rapid changes in treatment and respond quickly and accurately. However, there are still many challenges that need to be addressed. Finally, it will be important to understand the advantages and limitations of current drug therapies in clinical practice and new drug development to address these challenges.

## Figures and Tables

**Table 1 curroncol-32-00490-t001:** Phase II/III trials aimed at developing first-line drugs for HCC.

Trial Name (Registration No.)	Regimen	Control Arm	Phase	Ref.
IMbrave152	Atezolizumab + bevacizumab + tiragolumab	Atezolizumab + bevacizumab	III	[36]
TRIPLET-HCC	Atezolizumab + bevacizumab + ipilimumab	Atezolizumab + bevacizumab	III	[37]
NCT06109272	Livmoniplimab + budigalimab	Atezolizumab + bevacizumab/tremelimumab + durvalumab	II/III	[38]
SIERRA	Tremelimumab + durvalumab	None	IIIb	[39]
NCT04194775	Nofazinlimab + lenvatinib	Lenvatinib	III	[40]
NCT04720716	IBI310 + sintilimab	Sorafenib	III	[41]
APOLLO NCT04344158	Anlotinib + penpulimab	Sorafenib	III	[42]
NCT05408221	Rulonilimab + lenvatinib	Lenvatinib	II/III	[43]
NCT04401800	Tislelizumab + lenvatinib	None	II	[44]
RENOBATE (NCT04310709)	Regorafenib + nivolumab	None	II	[45]
NCT05924997	Adebrelimab + camrelizumab + apatinib	None	Ib/II	[46]

**Table 2 curroncol-32-00490-t002:** Phase II/III trials aimed at developing second-line drugs for HCC.

Trial Name (Registration No.)	Regimen (Control Arm)	Eligible Patients	Phase	Ref.
IMbrave251 (NCT04770896)	Atezolizumab + lenvatinib or sorafenib (lenvatinib or sorafenib)	Failure of treatment with atezolizumab and bevacizumab	III	[50]
LIVERATION (NCT05201404)	Namodenoson (placebo)	Child–Pugh B7 Disease progression after first-line therapy	III	[51]
NCT04696055	Regorafenib + pembrolizumab	After PD1/PD-L1 ICIs	II	[52]
FAITH (NCT06031480)	TQB2450 + anlotinib	Failure of prior ICIs	II	[53]
NCT05148195	Suvemcitug + envafolimab	Received at least one prior treatment	II	[54]
NCT05873244	Zabadinostat (CXD101) + geptanolimab (Lenvatinib or sorafenib)	After PD1/PD-L1 ICIs	II	[55]
jRCT1041200068	Atezolizumab + bevacizumab	After lenvatinib	II	[56]

PD1: Programmed cell death protein-1, PD-L1: Programmed cell death ligand-1, ICI: Immune checkpoint inhibitor.

**Table 3 curroncol-32-00490-t003:** Ongoing Phase II/III trials comparing TACE with TACE in combination with mTKIs and/or ICIs.

Trial Name (Registration No.)	Regimen	Control Arm	Phase	Ref.
ABC-HCC (NCT04803994)	TACE + additional on-demand TACE	Atezolizumab + bevacizumab	III	[68]
REPLACE (NCT04777851)	Regorafenib + pembrolizumab	TACE alone	III	[69]
IMPACT (jRCTs051230037)	Atezolizumab + bevacizumab additional TACE for stable disease patients	Atezolizumab + bevacizumab alone for stable disease patients	III	[74]
TALENTACE (NCT04712643)	Atezolizumab + bevacizumab + on-demand TACE	TACE alone	III	[75]
EMERALD-3 (NCT05301842)	TACE + tremelimumab + durvalumab ± lenvatinib	TACE alone	III	[76]
LEAP-012 (NCT04246177)	Lenvatinib + pembrolizumab + TACE	TACE alone	III	[77]
CheckMate74W (NCT04340193)	Nivolumab + ipilimumab + TACE	Nivolumab + TACE/TACE alone	III	[78]
NCT05003700	Camrelizumab + lenvatinib + RALOX-HAIC		II	[79]

TACE: Transarterial chemoembolization, RA-LOX-HAIC: raltitrexed + oxaliplatin–hepatic arterial infusion chemotherapy.

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
