# Peer review of "Advances in Systemic Therapy for Hepatocellular Carcinoma and Future Prospects"

_curroncol, 2025, doi:10.3390/curroncol32090490_

Round 1
Reviewer 1 Report
Comments and Suggestions for Authors
The authors report on the current status of systemic therapy for mostly advanced stage HCC, including also a bit the use of systemic therapies in earlier stages of HCC. The report is mostly accurate with respect to the current status but there is little information about truly future prospects beyond trials that are already ongoing for some time.
Specific comments:
-2. The Sharp-Trial was not published in 2017: please correct
-Cosmic-312-trial: the OS was not significant but the PFS was, so the statement of lack of efficacy is not truly correct
-page 7: Kudo’s study did not use “tendency-matched controls” but “propensity-matched controls”; please correct
-the more internationally relevant European HCC-guidelines (compared to ESMO, which included therapies of little importance for clinical practice) also published in 2025 are the EASL-HCC guidelines. I would suggest to include those (instead of or in addition to the ESMO-guidelines) into the discussion of current HCC-guidelines.
Author Response
Response to Reviewer 1
Thank you very much for your important comments. We initially decided to limit this review to studies with high level of evidence, including guidelines, and therefore did not mention Phase I and II trials. However, in light of your comments, we have reconsidered our position and decided to also discuss the future prospects for hepatocellular carcinoma treatment, including Phase II trials.
Page number 6, line 243-249
Page number 7, line 287-341
Page number12, line 469-473
Replies to individual comments
1.The Sharp-Trial was not published in 2017: please correct
→Thank you for pointing this out. We apologize for our mistake and the description has been corrected to 2008.
Page number 2, line 84
2.Cosmic-312-trial: the OS was not significant but the PFS was, so the statement of lack of efficacy is not truly correct
→ Thank you for pointing this out. We agree with this comment and have added text indicating that the PFS was significantly improved in the Cosmic-312 trial and that the trial is ongoing.
Page numbers5, lines 207-211
- -page 7: Kudo’s study did not use “tendency-matched controls” but “propensity-matched controls”; please correct
→Thank you very much for your comment. We have corrected the information accordingly.
Page number 11, line 422
-the more internationally relevant European HCC-guidelines (compared to ESMO, which included therapies of little importance for clinical practice) also published in 2025 are the EASL-HCC guidelines. I would suggest to include those (instead of or in addition to the ESMO-guidelines) into the discussion of current HCC-guidelines.
→Thank you for this very important suggestion. We have also referred to the EASL guidelines.
Page number97, lines 325-341

Reviewer 2 Report
Comments and Suggestions for Authors
There are some comments.
It would be better to define "advanced stage hepatocellular carcinoma (HCC)"
It would be better to describe whether there are differences in treatment efficacy based on the pathological characteristics of HCC (e.g., histological subtypes, grading).
It would be better to put the references in the last column on the right side of Tables 1 and 2, and explain the abbreviations below the Tables.
It would be better to add a conclusion section.
Comments on the Quality of English LanguagePlease check the spacing.
For example, line 44, 2009(1).-> 2009 (1).
line 50, Regorafenib(3) -> Regorafenib (3)
Author Response
Response to Reviewer 2
- It would be better to define "advanced stage hepatocellular carcinoma (HCC)"
→Thank you for pointing this out. The definition of advanced hepatocellular carcinoma used in this review has been provided as follows:
In this review, we defined advanced hepatocellular carcinoma as cases with BCLC stage B or C and Child-Pugh class A, for which curative treatments such as ablation therapy, TACE monotherapy, or resection are not expected to be beneficial, and for which systemic drug therapy is indicated.
Page number 2, lines73-77,
- It would be better to describe whether there are differences in treatment efficacy based on the pathological characteristics of HCC (e.g., histological subtypes, grading).
→ Thank you for this very important suggestion. We have included a section discussing differences in the efficacy of drug therapy based on histological subtypes and gene expression.
Page numbers 2, line71-72,
Page number 14 lines 517-539 (Section6)
- -It would be better to put the references in the last column on the right side of Tables 1 and 2, and explain the abbreviations below the Tables.
→Thank you very much for these comments. We have moved the references to the last column on the righthand side of Tables 1 ,2 and 3, and explained the abbreviations below the Tables.
Page number6,8,12-13, line304,477
-It would be better to add a conclusion section.
→Thank you for this very important suggestion. We have added a concluding section.
Page number 15, line560-567 (Section8)
Please check the spacing.
→Thank you very much. We have corrected all the spacing issues.
Reviewer 3 Report
Comments and Suggestions for Authors
The manuscript is generally well-structured and offers valuable insights into the evolving treatment landscape of hepatocellular carcinoma, including sequential tyrosine kinase inhibitor, immunotherapy-based combinations, and integration with local therapies. However, several aspects require major revisions to improve clarity, consistency, and overall readability:
- While Section 2 (Advances and prospects in systemic therapy for advanced-stage hepatocellular carcinoma) presents a thorough overview of systemic therapies and current guideline recommendations, the content remains largely descriptive. The authors are encouraged to synthesize this information by identifying overarching trends, comparing treatment strategies, and highlighting key messages that support clinical decision-making.
- The manuscript currently includes two sections labeled as “Section 2,” which may cause confusion. Please revise the section numbering to ensure a coherent and logically structured presentation.
- In introduction, while the review provides a comprehensive overview of recent advances in HCC treatment, it omits the role of complementary approaches such as immunotherapy and traditional Chinese medicine. Including a brief discussion of this perspective would improve the completeness of the review. Relevant studies such as PMID: 39583310 and PMID: 39455405 are useful and could be cited.
- Terms such as “immune hot,” “immune excluded,” and “immune cold” are introduced without clear definitions or supporting references. These terms should be defined upon first mention, with appropriate citations to clarify their immunological and clinical significance in HCC.
- Some abbreviations, such as “ABC,” are not explained when first introduced. The authors should carefully review the manuscript to ensure all abbreviations are properly defined at first use, and that the “Abbreviations” list is updated accordingly to maintain clarity and professionalism.
- Tables 1 and 2 are insufficiently referenced or explained in the main text. It is recommended that the authors explicitly cite these tables at relevant points and briefly summarize their key content to guide the reader.
Author Response
- While Section 2 (Advances and prospects in systemic therapy for advanced-stage hepatocellular carcinoma) presents a thorough overview of systemic therapies and current guideline recommendations, the content remains largely descriptive. The authors are encouraged to synthesize this information by identifying overarching trends, comparing treatment strategies, and highlighting key messages that support clinical decision-making.
→Thank you very much for your extremely important advice. We have included some discussion that we believe will be useful in supporting clinical treatment decisions by comparing individual treatment strategies.
Page number10, line 369-392
- The manuscript currently includes two sections labeled as “Section 2,” which may cause confusion. Please revise the section numbering to ensure a coherent and logically structured presentation.
→Thank you very much for your kind attention. This was an error on our part that we apologize for. The section number has now been corrected.
Page number 10, line394
- In introduction, while the review provides a comprehensive overview of recent advances in HCC treatment, it omits the role of complementary approaches such as immunotherapy and traditional Chinese medicine. Including a brief discussion of this perspective would improve the completeness of the review. Relevant studies such as PMID: 39583310 and PMID: 39455405 are useful and could be cited.
→Thank you very much for your valuable advice. Reprogramming macrophages for immunotherapy is a very interesting topic, and we thought it would be a useful reference, but we decided not to cite it in this review because the topic is too broad to cover. However, traditional Chinese medicine is a very important perspective, so we cited two papers related to traditional Chinese medicine in the section on postoperative adjuvant chemotherapy in Chapter 5.
Page number14, line 511-514
- Terms such as “immune hot,” “immune excluded,” and “immune cold” are introduced without clear definitions or supporting references. These terms should be defined upon first mention, with appropriate citations to clarify their immunological and clinical significance in HCC.
→Thank you very much for your important advice. We have clarified the definitions of “immune hot,” “immune excluded,” and “immune cold,” cited the relevant sources, and added explanations and comments.
Page number3, line 110-125
- Some abbreviations, such as “ABC,” are not explained when first introduced. The authors should carefully review the manuscript to ensure all abbreviations are properly defined at first use, and that the “Abbreviations” list is updated accordingly to maintain clarity and professionalism.
→Thank you for your kind advice. We have checked all abbreviations.
Page number3, line 91,103,
Page number4, line 146,
Page number11, line 449-450,
- Tables 1 and 2 are insufficiently referenced or explained in the main text. It is recommended that the authors explicitly cite these tables at relevant points and briefly summarize their key content to guide the reader.
→Thank you for pointing this out. We have been added a brief explanation to the text for all tests shown in the tables in the main text. References corresponding to these tests have been added at the end of each table. In addition, we have added a table showing a Phase II/III clinical trial for the development of a second-generation drug for the treatment of hepatocellular carcinoma.
Page number5-6, line228-248,
Page numbe7-8, line287-318,
table 2: Page number8, line302-304,
Page number12, line461-466, line469-473,
Table 1,page6
Table 3 page12-13
Round 2
Reviewer 2 Report
Comments and Suggestions for Authors
The manuscript has been well-revised.
Comments on the Quality of English LanguagePlease check English grammar and spelling.
For example, Sorfenib -> Sorafenib
Lenvatinb -> Lenvatinib
However it -> However, it
Author Response
Thank you for pointing this out. We apologise for our mistake. We have checked all drug names and text and corrected any spelling mistakes and errors.
Page3 line 126
Page4 line 179
Page5 line 189, line192, line218
Reviewer 3 Report
Comments and Suggestions for Authors
I have no other suggestions.
Author Response
Thank you very much for your insightful review.